# Novel Digital Technique to Quantify the Area and Volume of Cement Remaining and Enamel Removed after Fixed Multibracket Appliance Therapy Debonding: An In Vitro Study

**DOI:** 10.3390/jcm9041098

**Published:** 2020-04-12

**Authors:** Álvaro Zubizarreta-Macho, Martina Triduo, Jorge Alonso Pérez-Barquero, Clara Guinot Barona, Alberto Albaladejo Martínez

**Affiliations:** 1Department of Endodontics, Faculty of Health Sciences, Alfonso X El Sabio University, 28691 Madrid, Spain; amacho@uax.es; 2Department of Orthodontics, Faculty of Medicine and Dentistry, University of Salamanca, 37008 Salamanca, Spain; martina.triduo.f@gmail.com (M.T.); albertoalbaladejo@hotmail.com (A.A.M.); 3Department of Stomatology, Faculty of Medicine and Dentistry, University of Valencia, 46010 Valencia, Spain; 4Department of Dentistry, Faculty of Health Sciencies, University Cardenal Herrera-CEU, CEU Universities, 46115 Valencia, Spain; claraguinot@gmail.com

**Keywords:** orthodontics, digital impression, geomorphometry, alignment, cement remaining, enamel removed

## Abstract

The aim of this study was to construct a novel, repeatable, reproducible, and accurate measurement protocol for the area and volume of the remaining cement after removal of fixed multibracket appliances, the area and volume of remaining cement after cement removal, the area and volume of enamel removed after cement removal, and the volume of cement used to adhere fixed multibracket appliances. A total of 30 brackets were cemented and removed with over 30 extracted teeth embedded into three experimental models of epoxy resin. The models were scanned before and after bracket placement, bracket debonding, and polishing the remaining cement. The brackets were submitted to micro-computed tomography. The standard tessellation language digital files were aligned, segmented, and re-aligned using geomorphometric software. The digital measurement technique accuracy, repeatability, and reproducibility were analyzed using Gage R&R statistical analysis. The variability attributable to the area and volume measurement techniques of the total variability of the samples was 0.70% and 0.11% for repeatability, respectively, and 0.79% and 0.01% for reproducibility, respectively. The re-alignment procedure is a repeatable, reproducible, and accurate technique that can be used to measure the area and volume of the remaining cement after removal of fixed multibracket appliances, the area and volume of remaining cement after cement removal, the area and volume of enamel removed after cement removal, and the volume of cement used to adhere the fixed multibracket appliance.

## 1. Introduction

Fixed multibracket appliance therapy has been widely used in orthodontic treatments to improve adhesion in terms of materials and techniques [1]. The treatment prognosis and time of orthodontic treatment are directly related to the adhesion strength of the multibracket appliance fixed to the dental surface [2]. The adhesion technique used for fixed multibracket appliance therapy includes the use of 37% orthophosphoric acid to allow the dissolution of the enamel surface, obtaining a micro-porosity of around 5–50 µm that improves the bonding-agent penetration through these micro-porosities inside the enamel surface and enhances the micro-retention effect of the resin cement into the conditioned enamel [3,4,5]. This is why, Bertacci et al. recommended a single application of a stannous fluoride-containing toothpaste on eroded enamel to prevent acid-induced permeability on the enamel surface after acid agent application [6]. However, resin cement removal after fixed multibracket appliance therapy debonding can lead to cracks or tear-outs, compromising tooth health and integrity [7], dental plaque formation due to the micro-roughness increase, dental hypersensitivity due to accidental enamel removed during the resin cement removal [8], and dental stains caused by the staining of the remaining cement after inadequate resin cement removal [9]. The upper central incisors are the most commonly affected teeth, followed by the upper lateral incisors and the canines [10]. These painful and aesthetic complications related to fixed multibracket appliance therapy debonding can cause discomfort to the patients and may require additional therapeutic procedures; so, the factors that influence these adverse effects must be controlled. Odegaard et al. reported a different bond failure pattern between ceramic and metal brackets: at the enamel–adhesive interface, at the bracket–adhesive interface [11], or ceramic brackets with mechanical retention [12]. These may occur due to the higher bond strength between ceramic brackets and the adhesive, or between enamel and adhesive in the case of metal brackets, so mechanical retention brackets probably leave more cement than chemical retention brackets. Samruajbenjakul and Kukiattrakoon reported that metal brackets should have a clinical detachment resistance between 6 and 8 MPa [13]; however, higher detachment resistance values may require excessive force to remove the bracket, possibly damaging enamel [14].

Some authors analyzed the cement removal procedures for maintaining the enamel surface integrity and highlighted tungsten carbide burs at high or low speeds [1,15,16], laser [17], fiberglass drills [18], and ultrasound [15,19]. The final remaining cement is polished with a rubber cup and diamond paste or aluminum oxide paste [14,15]. However, the best option for removing the remaining cement is through multi-blade tungsten carbide burs at high speed with irrigation [1,15]. Most of the studies used microscopic measurement methods, which are useful for assessing the enamel micro-roughness after the cement removal, but are impossible to apply in clinical settings, because it is necessary to extract the teeth [20]. Other studies used scanning electron microscopy [1], profilometry [20], atomic force microscope [18], and optic coherence tomography technology [21] to measure the damage produced to the enamel surface after bracket removal. Digital procedures have been introduced to dental practice due to their accuracy, simplicity, and versatility in acquiring information about dental treatments for diagnostic purposes, treatment planning, prosthesis and appliance fabrication, and for research [22]. The intraoral scans generate a standard tessellation language (STL) digital file by means of a cloud of points that create a tessella network, representing three-dimensional objects as polygons composed of tessellas of equilateral triangles [23,24]. The accuracy of the object is given by the tessella mesh density, allowing a precision range of ±10 μm.

The aim of this work was to describe a novel digital technique that could be used to quantify the amount of cement necessary to adhere fixed orthodontic therapy, the cement remaining after orthodontic treatment removal, and the enamel removed during cement removal, with a null hypothesis (H0) stating that no difference exists between the geomorphometric measurement protocol used with regards to the measurement accuracy of the amount of cement remaining after fixed multibracket appliance removal, the amount of orthodontic cement necessary to fix multibracket appliances, and the amount of enamel removed after fixed multibracket appliance removal.

## 2. Materials and Methods

### 2.1. Study Design

Thirty upper teeth representatives of all dental sectors, extracted for periodontal and orthodontic reasons, without caries, restorations, or fractures, were selected in this study at the Alfonso X El Sabio University (Madrid, Spain), Master Degree in Orthodontics at University of Salamanca (Salamanca, Spain), and the Department of Stomatology at University of Valencia (Valencia, Spain), between November 2019 and February 2020. A randomized controlled in vitro study was conducted in accordance with the principles defined in the German Ethics Committee’s statement for the use of organic tissues in medical research (Zentrale Ethikkommission, 2003), and was authorized by the Ethical Committee of the Faculty of Health Sciences, University Alfonso X El Sabio (Madrid, Spain), in December 2019 (process no. 03/2019). All patients provided informed consent to transfer the teeth for the study.

### 2.2. Experimental Procedure

The teeth were randomly (Epidat 4.1, Galicia, Spain) embedded into three experimental models of epoxy resin (ref.: 20-8130-128. EpoxiCure^®^, Buehler, IL, USA) with 14 teeth each (Figure 1a). The experimental models were submitted to a baseline intraoral scan (STL1; True Definition, 3M ESPE ™, Saint Paul, MN, USA) (by means of a 3D in-motion video imaging technology (Figure 2a). The images were captured following the manufacturer’s recommendations by first scanning the occlusal plane, followed by the vestibular and palatal faces. Later, the fixed multibracket appliance was cemented only on teeth 1.5–2.5 in the center of the buccal surface of the clinical crown with a photo-polymerized composite resin cement (Transbond™ XT, 3M ESPE ™, Saint Paul, MN, USA) before etching the enamel buccal surface with 37% orthophosphoric acid (VOCOCID, VOCO GmbH, Cuxhaven, Germany) for 20 s and photo-polymerized resin adhesive primer application (Unitek Transbond™ XT, 3M ESPE™, Saint Paul, MN, USA) for 20 s (Figure 1b). Finally, a post-cementation intraoral scan (STL2; True Definition, 3M ESPE™, Saint Paul, MN, USA) was performed (Figure 2b). Then, the fixed multibracket appliance was removed from teeth 1.5 to 2.5 with a specific instrument to remove the fixed multibracket appliance (MBT, 0.022, Pacific Orthodontics, Guadalajara, Spain) (Figure 1c) and a post-removing bracket intraoral scan (STL3; True Definition, 3M ESPE™, Saint Paul, MN, USA) was performed (Figure 2c). The remains of composite resin cement were gently removed with a unidirectional movement by a single operator, using a low-speed contra angle handpiece, (W&H WE-99 LED G, Bürmoos, Austria) at 1500 rpm with profuse irrigation. A polishing diamond bur surface (ref. 882 314 012, Komet Medical, Lemgo, Germany) was used for each experimental model, until there was no cement clinically visible (Figure 1d) and a post-removing cement intraoral scan (STL4; True Definition, 3M ESPE™) was performed (Figure 2d). The brackets were submitted to a micro-computed tomography scan (micro-CT; STL5; Skyscan 1176, Bruker-MicroCT, Kontich, Belgium) with the following exposure parameters: 160.0 kilovolt peak, 56.0–58.0 microamperes, 500.0 ms, 720 projections 4 frames, a tungsten target between 0.25 and 0.375 mm, a 3 µm resolution, and a pixel size of 0.127 µm, to obtain accurate STL digital files of the surface of each of the fixed multibracket appliances.

### 2.3. Alignment Procedure

Once STL1–4 were imported to reverse engineering geomorphometric software (3D Geomagic Capture Wrap, 3D Systems^©^, Rock Hill, SC, USA); a full-arch alignment procedure was conducted. STL1 was considered the reference digital file, and STL2–4 were superimposed on it using the palatal surfaces of the anterior teeth and the occlusal and palatal surfaces of the posterior teeth, with the best fit algorithm. Afterward, the teeth from 15 to 25 of all STL files were segmented (Figure 3a–d) and individually three-dimensionally compared using the alignment of the STL files 2 (Figure 4a), 3 (Figure 4b), and 4 (Figure 4c) with STL1 used as the reference.

The spectrum was set to ±100 μm and the tolerance to ±10 μm (Figure 4d).

Then, a new alignment procedure (re-alignment) was performed. This re-alignment was individually performed from the previously segmented teeth 1.5–2.5 using the intact palatal surface of each tooth as reference to enable the re-alignment. The previously segmented teeth of STL1 were considered the reference and the corresponding segmented teeth of STL2–4 were superimposed, so the 3D position of the teeth of the STL1 was not modified in the process of re-alignment.

After the re-alignment, a 3D comparison was performed with the same spectrum and tolerance values previously described (Figure 5).

### 2.4. Measurement Procedure

After the alignment and re-alignment procedures, the following variables were measured: area and volume of the remaining cement after fixed multibracket appliance therapy removal, area and volume of the remaining cement after cement removal, area and volume of the enamel removed after cement removal, and the volume of the cement used to adhere the fixed multibracket appliance.

A random (Epidat 4.1, Galicia, Spain) tooth was selected and all the above-mentioned measures were calculated. Segmented tooth 2.3 of STL1 (Figure 6a), STL2 (Figure 6b), STL3 (Figure 6c), and STL4 (Figure 6d) aligned and re-aligned and the STL of the corresponding bracket obtained from the micro-CT (Figure 6e) were used to analyze the variables previously described.

STL1 (Figure 7a) and STL3 (Figure 7b) digital files of the previously segmented and re-aligned tooth 2.3 were selected to analyze the area and volume of the remaining cement after the fixed multibracket appliance therapy removal. The STL1 (Figure 7c) digital file of the tooth 2.3 was slightly overcontoured regarding STL3 (Figure 7d), which allowed us to differentiate the boundaries between both STL digital files. Afterward, the STL1 digital file recovered its original size and the area of the remaining cement was determined by comparing the boundaries of STL1 (Figure 7e) and STL3 (Figure 7f) digital files. The remaining cement area was selected and isolated in STL1 (Figure 7g) and STL3 (Figure 7h) digital files by reverse selection, and the normals of the tessella network of the selected area of the remaining cement of the STL1 (Figure 7g) digital file were flipped to obtain a closed polygon with the selected area of the remaining cement in the STL3 digital file (Figure 7h). This allowed us to measure the remaining cement after fixed multibracket appliance removal (Figure 7i).

The isolated area of the remaining cement in STL3 (Figure 8a) and STL4 (Figure 8b) digital files of the previously segmented and re-aligned tooth 2.3 were selected to analyze the area and volume of the remaining cement after removal using an intersection Boolean operation calculated by 3D Geomagic Capture Wrap software 2017.0.0. (3D Systems^©^, Rock Hill, Rock Hill, SC, USA) (Figure 8c).

The STL1 (Figure 9a) and STL4 (Figure 9b) digital files of the previously segmented and re-aligned tooth 2.3 were selected to analyze the area and volume of the enamel removed after cement removal. The measurement procedure (Figure 9a–i) was performed following the measurement procedure previously described to determine the area and volume of the remaining cement after fixed multibracket appliance therapy removal.

Finally, the STL1, STL2 (Figure 10a), and STL5 (Figure 10c) digital files of the previously segmented and re-aligned tooth 2.3 were selected to analyze the volume of the cement used to adhere the fixed multibracket appliance. The STL2 digital file was transformed into a solid polygon (Figure 10b) and the bracket of the STL2 digital file was removed using a subtractive Boolean operation with the STL5 digital file (Figure 10c). A new solid polygon was obtained with the volume of the cement used to adhere the fixed multibracket appliance over the STL2 digital file (Figure 10d).

The STL1 (Figure 11a) and new solid polygon of the STL2 (Figure 11b) digital files of the previously segmented and re-aligned tooth 2.3 were selected to analyze the volume of the cement used to adhere the fixed multibracket appliance. The measurement procedure (Figure 11a–i) was performed following the measurement procedure previously described to determine the area and volume of the remaining cement after removal of the fixed multibracket appliance and the volume of cement used to adhere the fixed multibracket appliance (Figure 11i).

### 2.5. Validation of the Repeatability and Reproducibility

To validate the repeatability of this new protocol, the measurements described above were calculated six times using the same operator (Operator A). The measurements were calculated six times by another operator (Operator B) to validate the reproducibility of this new measurement technique.

### 2.6. Statistical Analysis

Statistical analysis of the measurement variables was conducted using SAS 9.4 (SAS Institute Inc., Cary, NC, USA). Descriptive statistics are expressed as mean and SD for quantitative variables. Comparative analysis was performed by comparing the mean deviation values between the STL digital files in area and volume using Student’s *t*-test as the variables had a normal distribution. The statistical significance was set to *p* ˂ 0.05. Gage R&R statistical analysis was conducted to analyze the repeatability and reproducibility of this measurement technique.

## 3. Results

The means and SD values for aligned and re-aligned accuracies of the STL digital files are displayed in Table 1 and Figure 12.

Student’s *t*-test revealed statistically significant differences between the aligned and re-aligned values of the STL digital files (*p* ˂ 0.000; Table 1, Figure 12a) after analyzing three measures in 10 teeth (Figure 12b). We observed a 25.13% increase in the tessella network of the re-aligned STL digital files within the pre-established ±10 µm tolerance range with a power of sample size of 100%.

Table 2 and Figure 13 display the means and SD values necessary to analyze the repeatability of the measurement technique for the area and volume of the remaining cement after fixed multibracket appliance therapy removal, area and volume of the remaining cement after cement removal, area and volume of the enamel removed after cement removal, and the volume of the cement necessary to adhere the fixed multibracket appliance.

The Gage R&R statistical analysis of the measurement technique in terms of the area and volume of the remaining cement after fixed multibracket appliance removal, area and volume of the remaining cement after cement removal, area and volume of the enamel removed after cement removal, and the volume of the cement used to adhere the fixed multibracket appliance showed that the variabilities attributable to the area and volume measurement techniques were 0.70% and 0.11%, respectively, of the total variability of the samples. The technique demonstrated a high repeatability for the area and volume measurement techniques and did not show statistically significant differences between the analyzed means of the area (*p* = 1.000; Figure 13a) and volume (*p* = 1.000; Figure 13b).

Table 3 and Figure 14 display the means and SD values necessary to analyze the reproducibility of the measurement technique for the area and volume of the remaining cement after the fixed multibracket appliance therapy removing, area (µm^2^) and volume (µm^3^) of the remaining cement after the cement removing, area (µm^2^) and volume (µm^3^) of the enamel removed after the cement removing and the volume (µm^3^) of the cement used to adhere the fixed multibracket appliance.

The Gage R&R statistical analysis of the measurement technique of the area and volume of remaining cement after fixed multibracket appliance therapy removal, area and volume of the remaining cement after cement removal, area and volume of enamel removed after cement removal, and the volume of the cement used to adhere the fixed multibracket appliance performed by the two operators showed that the variability attributable to the area and volume measurement techniques was only 0.79% and 0.01% of the total variability of the samples, respectively. The technique demonstrated a high reproducibility for area and volume measurement techniques and did not show statistically significant differences between the means of the area (*p* = 0.785; Figure 14a) and volume (*p* = 0.951; Figure 14b).

## 4. Discussion

The results obtained in the present study rejected the null hypothesis (H0) of no difference between the geomorphometric measurement protocol used for the area and volume of the remaining cement after fixed multibracket appliance therapy removal, area and volume of the remaining cement after cement removal, area and volume of enamel removed after cement removal, and the volume of cement used to adhere the fixed multibracket appliance.

Microscopic visual measurement techniques were used to analyze the enamel surface damage after fixed multibracket appliance therapy removal using indexes created ad hoc: enamel surface index (ESI) [25], enamel damage index (EDI) [26], line angle grooves (LAG) [27], or enamel surface rating system [3]. SEM allows the identification of the remaining cement on the enamel surface after fixed multibracket appliance therapy removal [16] and the rugosimetry devices incorporated with SEM allow profilometric analysis of enamel surface orography [20]. However, none of these techniques measure the volumes of cement and enamel removed and require tooth extraction [28]. Additional measurement techniques were described for analyzing the enamel surface alterations after fixed multibracket appliance therapy removal: fluorescence [28], atomic force microscope [18], and optic coherence tomography [22]. The fluorescence measurement visual technique has been widely used to differentiate composite from enamel and for identifying the remaining cement after fixed multibracket appliance therapy removal, due to the luminescent differences between composite resin and tooth structures, but cannot measure the volumes of cement and enamel removed [29,30]. Macroscopic visual techniques have been used to identify enamel surface damage and tooth discolorations related to the remaining cement after fixed multibracket appliance therapy removal [31]. These visual techniques allow clinical measurements in clinical settings, but area and volume cannot be measured. Briefly, no measurement technique is available for the clinical setting that allows repeatable, reproducible, and acute measurement for analyzing the area and volume of the remaining cement after fixed multibracket appliance therapy removal, the area and volume of remaining cement after cement removal, area and volume of enamel removed after cement removal, and the volume of the cement used to adhere fixed multibracket appliances. For this reason, the geomorphometric analysis of the STL digital files obtained from intraoral scans was used to analyze all these parameters. However, some studies highlighted the factors related to the inaccuracy of digital impression systems. Jivanescu et al. reported that the presence of adjacent teeth can decrease the view of interproximal surfaces [32]. Ender et al. reported no statistically significant differences between the precision values of Cerec Bluecam, Cerec Omnicam, Itero, Lava C.O.S., Lava True Definition, TRIOS, and TRIOS color digital impression systems for dental nature whole-arch scanning [33]. However, Kuhr et al. reported statistically significant differences between the trueness values of Lava True Definition (23.0 µm) and Cerec Omnicam (214 µm) but no statistically significant differences between the trueness values of Lava True Definition and TRIOS (37.0 µm) for dental nature whole-arch scanning [34]. Guth et al. reported that Lava True Definition offers higher levels of trueness (21.8 µm) than Cerec Bluecam (34.2 µm), Cerec Omnicam (43.3 µm), Itero (49.0 µm), Lava C.O.S. (47.7 µm), TRIOS (25.7 µm), and TRIOS color (26.1 µm) digital impression systems for dental nature partial-arch scanning [35]. These trueness values obtained from the scanning of dental nature partial arch are slightly lower than those obtained from dental nature whole arch, potentially due to the higher tessella network of the whole-arch STL digital files [36]. Powder-dependent digital impression systems provide more accurate STL digital files than non-powder-dependent digital impression systems on different scanned substrates [10,37], and specifically on metal abutments [7], because powder prevents light refraction and scattering, allowing an accurate determination of the object depth [38]. The imprecision associated with intraoral digital impression systems can lead to biological and mechanical problems such as caries, prosthesis misfit, and loss of prosthesis retention [39]. Imprecise surface details prevent precise articulation and occlusion establishment [40]. Several factors may affect the digital impression accuracy. One of these factors is the distance from the first point scanned, used as a reference, and the succeeding points in the scanned structures, which would have been stitched to the previous one. Each individual stitch represents a chance of incurring an error, preventing the correct alignment of the full-arch STL digital files with reverse engineering geomorphometric software, and requiring the segmentation and re-alignment of the individualized STL digital files, obtaining a 25.13% larger tessella network at the re-aligned STL digital files within the pre-established ±10 µm tolerance range. This increase in the tessella network inside the tolerance range was observed in 100% of the re-aligned STL digital files, which was more evident in the re-aligned STL4 digital file.

Metal brackets were selected in this study because mechanic retention bracket systems present a bond failure pattern at the bracket–adhesive interface [2], leaving residual cement after the fixed multibracket appliance therapy removal. In this situation, the area and volume of remaining cement after fixed multibracket appliance therapy removal, area and volume of remaining cement after cement removal, area and volume of the enamel removed after cement removal, and the volume of cement used to adhere the fixed multibracket appliance must be analyzed. To determine the accuracy, repeatability, and reproducibility of the digital measurement technique, Gage R&R statistical analysis was performed. This method was reported to be useful for evaluating the precision and consistency of a measurement process [41]. We determined how much of the variability in the process is due to variation in the measurements system; we used inference techniques to estimate repeatability and reproducibility. When a measurement process is conducted, the total process variation consists of part-to-part variation plus measurement system variation. Measurement system variation is determined by the repeatability, which is described as the variability of the measures performed by the same operator when the same part is measured, and the reproducibility, which is the variability of the measures performed by different operators when the same part is measured. Ideally, very little of the variability should be due to repeatability and reproducibility. Differences between parts (part-to-part) should account for most of the variability. When variability occurs, the measurement system can reliably distinguish between parts. The Gage R&R results of the area and volume measurements for the repeatability (0.70% and 0.11%, respectively) and the reproducibility (0.79% and 0.01%, respectively) are <1%, which are considered acceptable for a measurement system. From >10% to <30% is considered acceptable depending on the application, cost, or other factors, and at >30%, the measurement system is considered unacceptable and should be improved [42].

Calculating the cement used to adhere a multibracket appliance is considered possible for future investigations where digital planning is conducted and indirect bonding is performed. With this new protocol, the volume of cement planned for use and the real amount of cement used can be measured.

The findings show that the geomorphometric re-aligned measurement protocol provides a more accurate digital measure technique for the area and volume of the remaining cement after fixed multibracket appliance therapy removal, area and volume of remaining cement after cement removal, area and volume of enamel removed after cement removal, and the volume of cement used to adhere the fixed multibracket appliance. Further research is needed to determine the influence of fixed multibracket appliance removal on potential clinical complications.

## 5. Conclusions

In conclusion, within the limitations of this study, the results showed that the re-aligned measurement protocol is a repeatable, reproducible, and acute measure technique for the area and volume of remaining cement after fixed multibracket appliance therapy removal, area and volume of remaining cement after cement removal, area and volume of enamel removed after cement removal, and the volume of cement used to adhere the fixed multibracket appliance. Clinical trials are necessary to analyze the potential clinical complications associated with fixed multibracket appliance removal.

## Figures and Tables

**Figure 1 jcm-09-01098-f001:**
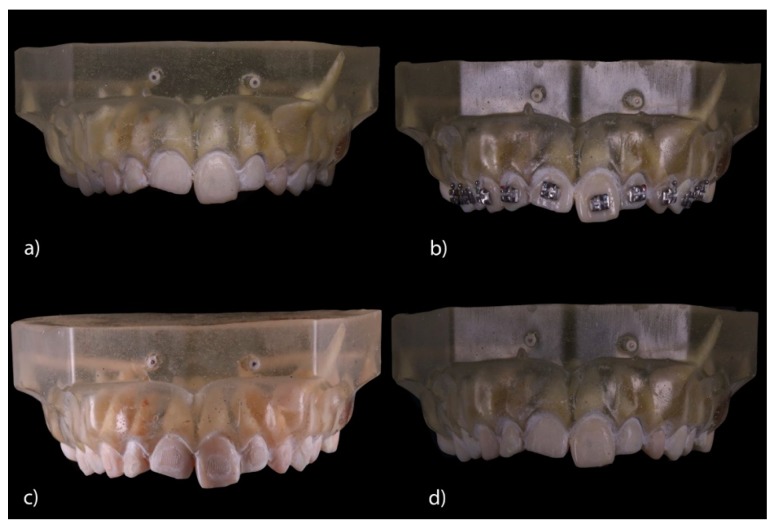
(**a**) Frontal view of the experimental model of extracted teeth; (**b**) experimental model with cemented fixed multibracket appliance; (**c**) experimental model with fixed multibracket appliance removed; and (**d**) experimental model with cement removed.

**Figure 2 jcm-09-01098-f002:**
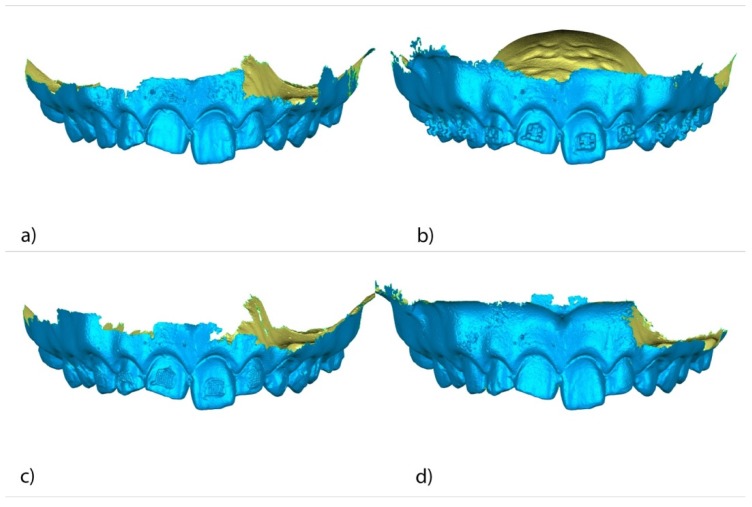
(**a**) Frontal view of the standard tessellation language (STL)1, (**b**) STL2, (**c**) STL3, and (**d**) STL4.

**Figure 3 jcm-09-01098-f003:**
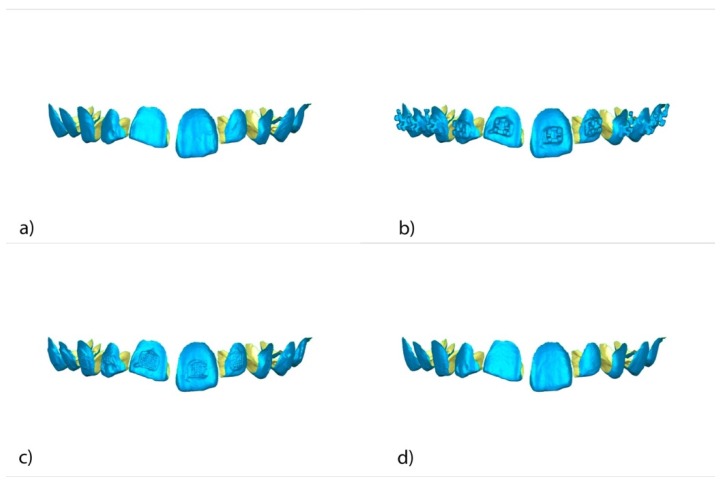
Frontal view of the individually segmented (**a**) STL1, (**b**) STL2, (**c**) STL3, and (**d**) STL4.

**Figure 4 jcm-09-01098-f004:**
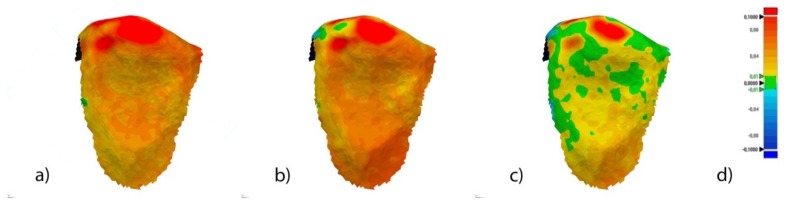
Palatal view of the three-dimensional comparison of the alignment of the segmented tooth 2.3 between (**a**) STL1 and STL2, (**b**) STL1 and STL3, and (**c**) STL1 and STL4. (**d**) Spectrum values used in (**a**–**c**). Warm colors represent a volume increase, cold colors represent a volume decrease, and green represents an accurate alignment.

**Figure 5 jcm-09-01098-f005:**
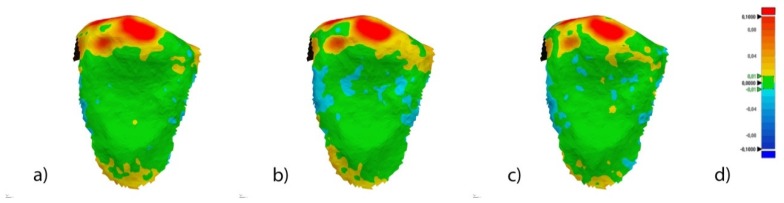
Palatal view of the three-dimensional comparison of the re-alignment of segmented tooth 2.3 between (**a**) STL1 and STL2, (**b**) STL1 and STL3, and (**c**) STL1 and STL4. (**d**) Spectrum values used in Figure 4a–c. Warm colors represent a volume increase, cold colors represent a volume decrease, and green represents an accurate alignment.

**Figure 6 jcm-09-01098-f006:**
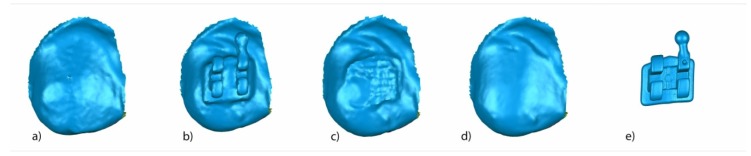
Buccal view of segmented tooth 2.3 of (**a**) STL1, (**b**) STL2, (**c**) STL3, and (**d**) STL4. (**e**) STL of the bracket 2.3 obtained from the micro-computed tomography.

**Figure 7 jcm-09-01098-f007:**
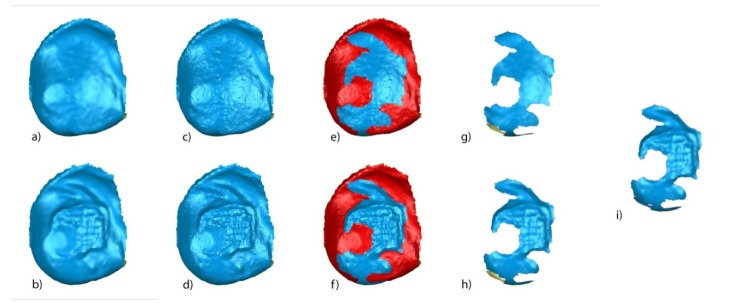
(**a**) Buccal view of segmented tooth 2.3 of STL1 and (**b**) STL3; slight overcontouring of (**c**) STL1 and (**d**) STL3; (**e**) boundary of the volumetric comparison between STL1 and STL3 over STL1; (**f**) boundary of the volumetric excess comparison between STL1 and STL3 over STL3; (**g**) new mesh with the remaining cement over the STL1 digital file; (**h**) new mesh with the remaining cement over the STL1 digital file; and (**i**) combined meshes with the remaining cement after bracket removal of STL1 and STL3, obtaining a closed polygon.

**Figure 8 jcm-09-01098-f008:**
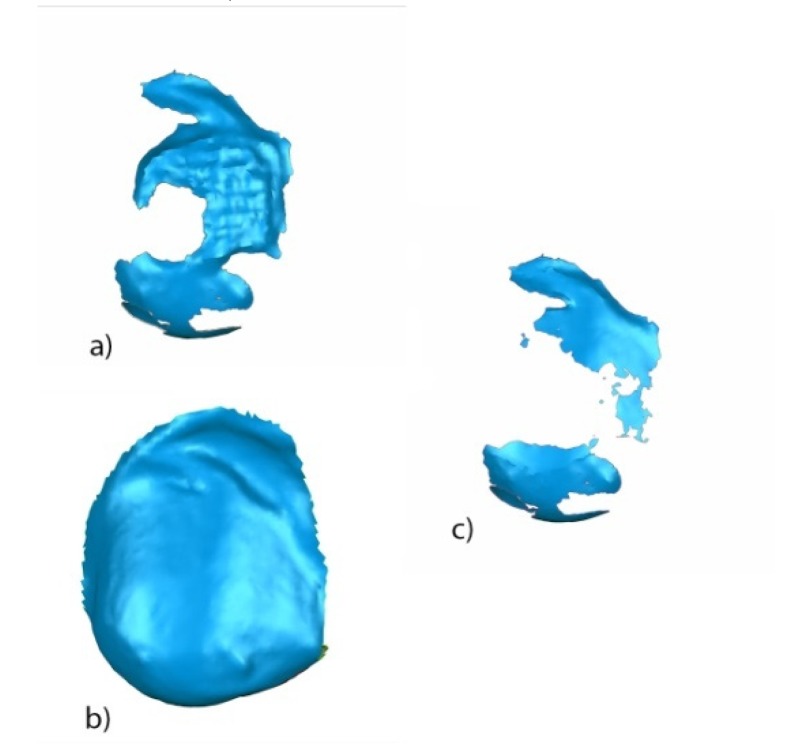
Mesh of the remaining cement after bracket removal in (**a**) STL3 and (**b**) STL4, and (**c**) excess remaining cement after removal in STL4.

**Figure 9 jcm-09-01098-f009:**
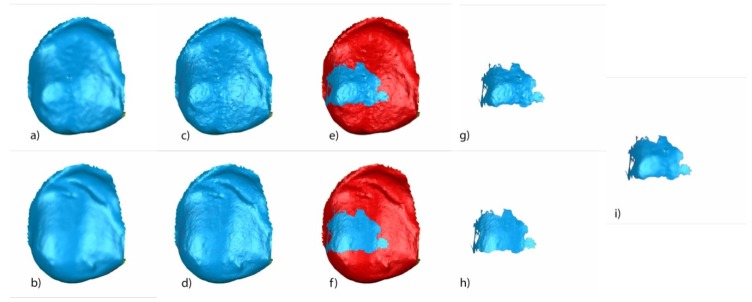
Buccal view of the segmented tooth 2.3 of (**a**) STL1 and (**b**) STL4; slight overcontouring of (**c**) STL1 and (**d**) STL4; (**e**) boundary of the volumetric comparison between STL1 and STL4 over STL1; (**f**) boundary of the volumetric defect comparison between STL1 and STL4 over STL4; new mesh with the enamel removed over the (**g**) STL1 and (**h**) STL4 digital files; and (**i**) combined meshes with the enamel removed of STL1 and STL4, obtaining a closed polygon.

**Figure 10 jcm-09-01098-f010:**
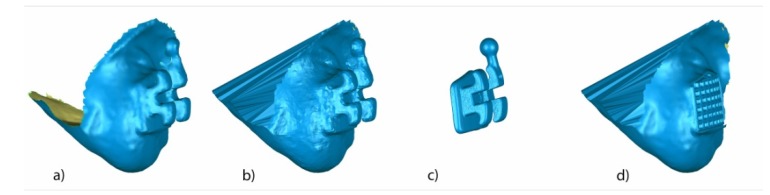
Proximal view of (**a**) the segmented tooth 2.3 of STL2 and (**b**) the solid mesh of tooth 2.3 of STL2. The mesh is not only in the perimeter, the inside is also filled. (**c**) STL5 and (**d**) solid mesh of tooth 2.3 of STL2 with the volume of the cement used to adhere and without the bracket.

**Figure 11 jcm-09-01098-f011:**
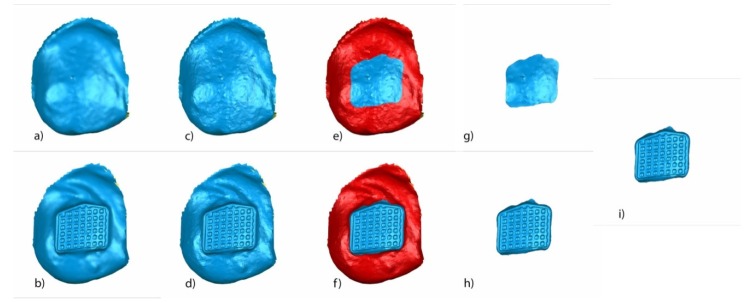
(**a**) Buccal view of segmented tooth 2.3 of STL1; (**b**) STL2 without bracket; slight overcontouring of (**c**) STL1 and (**d**) STL2 without bracket; boundary of the volumetric comparison between (**e**) STL1 and STL2 without bracket over STL1 and (**f**) STL1 and STL2 without bracket over STL2 without bracket; new mesh with the volume of the cement used over the (**g**) STL1 digital file and (**h**) STL2 without bracket digital file; and (**i**) combined meshes with the volume of cement used for STL1 and STL2 without bracket, obtaining a closed polygon.

**Figure 12 jcm-09-01098-f012:**
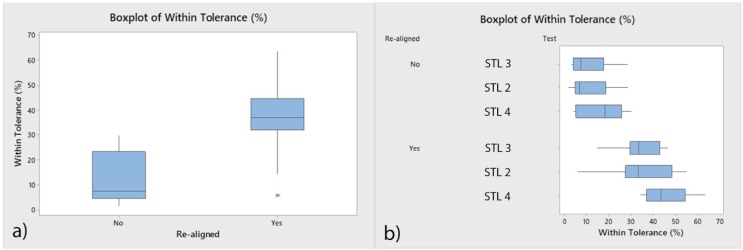
(**a**) Box plot of the total mean and standard deviation (SD) values and (**b**) mean and SD values of the aligned and re-aligned samples of each measurement.

**Figure 13 jcm-09-01098-f013:**
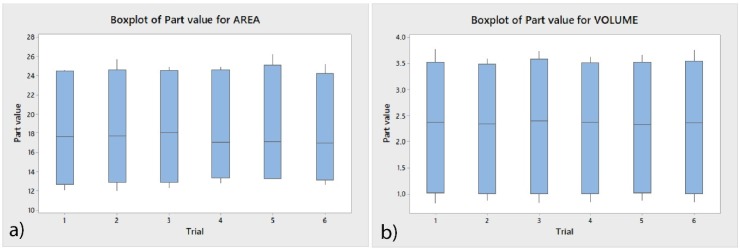
(**a**) Box plot of the mean and SD values of the area and (**b**) volume measured after the re-alignment of the STL digital files.

**Figure 14 jcm-09-01098-f014:**
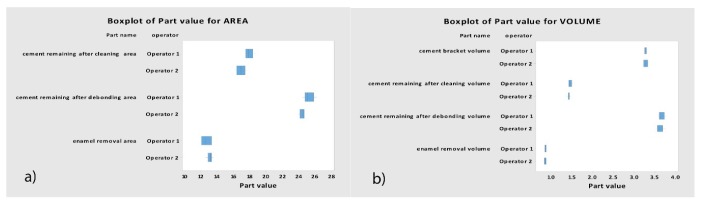
(**a**) Box plot of the mean and SD values of the area and (**b**) volume measured after the re-alignment of the STL digital files between two operators.

**Table 1 jcm-09-01098-t001:** Descriptive statistics of the aligned accuracy of the standard tessellation language (STL) digital files.

	*n*	Mean	SD	Minimum	Maximum
Aligned (µm)	30	12.75 ^a^	9.80	1.58	29.71
Re-aligned (µm)	30	37.88 ^b^	12.22	5.62	63.41

SD: standard deviation. ^a,b^ Statistically significant differences between groups (*p* < 0.05).

**Table 2 jcm-09-01098-t002:** Descriptive statistics of the measurement variables.

Variable	*n*	Mean	SD	Minimum	Maximum
Area of remaining cement after bracket removal (µm^2^)	12	24.8030	0.6340	23.8450	26.1720
Volume of remaining cement after bracket removal (µm^3^)	12	3.6417	0.0761	3.5398	3.7745
Area of remaining cement after cement removal (µm^2^)	12	17.4010	0.7650	16.1370	18.7650
Volume of remaining cement after cement removal (µm^3^)	12	1.4392	0.0351	1.3766	1.5138
Area of enamel removed after cement removal (µm^2^)	12	12.8390	0.5110	11.9900	13.4810
Volume of enamel removed after cement removal (µm^3^)	12	0.8576	0.0211	0.8114	0.8844
Volume of cement used to adhere brackets (µm^3^)	12	3.2794	0.0366	3.2146	3.3242

**Table 3 jcm-09-01098-t003:** Descriptive statistics of the measurement variables between operators.

Operator		*n*	Mean	SD	Minimum	Maximum
A	Area of remaining cement after bracket removal (µm^2^)	6	25.208	0.631	24.410	26.172
B	Area of remaining cement after bracket removal (µm^2^)	6	24.397	0.303	23.845	24.678
A	Volume of remaining cement after bracket removal (µm^3^)	6	3.6620	0.0742	3.5840	3.7745
B	Volume of remaining cement after bracket removal (µm^3^)	6	3.6213	0.0789	3.5398	3.7600
A	Area of remaining cement after cement removal (µm^2^)	6	17.929	0.519	17.361	18.765
B	Area of remaining cement after cement removal (µm^2^)	6	16.873	0.591	16.137	17.676
A	Volume of remaining cement after cement removal (µm^3^)	6	1.4564	0.0353	1.4221	1.5138
B	Volume of remaining cement after cement removal (µm^3^)	6	1.4220	0.0276	1.3766	1.4605
A	Area of enamel removed after cement removal (µm^2^)	6	12.606	0.588	11.990	13.274
B	Area of enamel removed after cement removal (µm^2^)	6	13.071	0.314	12.563	13.481
A	Volume of enamel removed after cement removal (µm^3^)	6	0.8604	0.0163	0.8318	0.8757
B	Volume of enamel removed after cement removal (µm^3^)	6	0.8547	0.0263	0.8114	0.8844
A	Volume of cement used to adhere brackets (µm^3^)	6	3.2823	0.0271	3.2397	3.3175
B	Volume of cement used to adhere brackets (µm^3^)	6	3.2765	0.0468	3.2146	3.3242

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
