# Peer review of "Novel Digital Technique to Quantify the Area and Volume of Cement Remaining and Enamel Removed after Fixed Multibracket Appliance Therapy Debonding: An In Vitro Study"

_jcm, 2020, doi:10.3390/jcm9041098_

Round 1
Reviewer 1 Report
Summary
The study showed a new protocol for quantifying the amount of cement necessary for adhesion to fixed orthodontic therapy, the cement remaining after orthodontic treatment removing and the enamel removed during the cement removing. They have compared the STL of the experimental models of epoxy resin containing 14 teeth extracted each, before and after brackets placement, after brackets debonding and after polish cement remaining. They have also checked the repeatability and reproducibility of this measurement technique. The manuscript is well-written and interesting to the community. Despite, I have some minor concerns:
Major issues: Results, Discussion, and conclusion
I am not sure if it is possible to quantify the amount of cement necessary to adhere to fixed orthodontic therapy. The data obtained show the volume of the cement under the fixed multibracket appliance. How do you know it is the amount of cement necessary to adhere to fixed orthodontic therapy in terms of volume?
So I think the results do not support this hypothesis.
Also, It could have been better if there were some information about the limitations of the study and areas for future work.
Minor issues:
Line 102 in this period the authors reported three experimental models 102 of epoxy resin with 14 teeth each, for a total of 42 teeth but the sample was composed of 30 teeth, please clarify.
Please, check typographic errors and improve punctuation.
Author Response
Dear Reviewer 1,
I’m pleased to resubmit the manuscript of the work entitled, “Novel digital technique to quantify the area and volume of cement remaining and enamel removed after fixed multibracket appliance therapy debonding: An in vitro study”.
I will respond to your comments and suggestions below:
Reviewer 1: English language and style are fine/minor spell check required.
Response: In order to adapt to the reviewer 1 comments, we have sent the manuscript to the English editing service of MDPI editorial (We attached the English Editing Certificate).
Reviewer 1: I am not sure if it is possible to quantify the amount of cement necessary to adhere to fixed orthodontic therapy. The data obtained show the volume of the cement under the fixed multibracket appliance. How do you know it is the amount of cement necessary to adhere to fixed orthodontic therapy in terms of volume? So I think the results do not support this hypothesis. Also, It could have been better if there were some information about the limitations of the study and areas for future work.
Response: In order to respond to the reviewer 1 comments, we have change the “necessary cement” for “cement used” in all the manuscript (figures included) as it is true that we don’t know if it is the exactly the amount of cement necessary, but for sure it is the cement used.
In addition we have written in discussion the importance of this capability of measure the volume of cement used: “The possibility of calculate the cement used to adhere the multibracket appliance is considered interesting for future investigations, where a digital planning is done and an indirect bonding is performed. It is possible with this new protocol, to measure the volume of cement we planned to used and the real amount of cement used.”
Reviewer 1: Line 102 in this period the authors reported three experimental models 102 of epoxy resin with 14 teeth each, for a total of 42 teeth but the sample was composed of 30 teeth, please clarify. Please, check typographic errors and improve punctuation.
Response: In order to respond to the reviewer 1 comments, we clarify that 3 experimental models with 14 teeth each (from tooth 1.7 to tooth 2.7) were manufactured, but in the study the brackets were cemented from the tooth 1.5 to the tooth 2.5 (it is indicated in line 111).
We take this opportunity to thank the recommendations and suggestions made by you to improve the document.
Yours sincerely,
Reviewer 2 Report
The authors performed an extensive digital work to understand modifications on enamel surface and remaining cement after debonding. There are minor corrections to be performed form methodological point of view, what I do prefer is to make the article easier to understand: I found it very difficult to read and understand and in some points it results redundant.
English need major revision
Abstract: not clear. Please rephrase. In the introduction and inmate conclusions.
Material and methods:
30 teeth or 42? Is not clear.
No mention about integrity of selected teeth was underlined.
Who removed the remain composite from teeth? Which handpiece? With or without water? Speed? One bur for all teeth? Same operator? Which force applied?
It's not clear why 2 alignments were required.
It's not clear how many teeth were evaluated in total (all the teeth?).
Why a diamond bur was used to eliminate the cement? Literature reports other kind of burs for this procedure.
Validation: any test was performed?
Author Response
Dear Reviewer 2,
I’m pleased to resubmit the manuscript of the work entitled, “Novel digital technique to quantify the area and volume of cement remaining and enamel removed after fixed multibracket appliance therapy debonding: An in vitro study”.
I will respond to your comments and suggestions below:
Reviewer 2: English need major revision.
Response: In order to adapt to the reviewer 2 comments, we have sent the manuscript to the English editing service of MDPI editorial (We attached the English Editing Certificate).
Reviewer 2: Abstract: not clear. Please rephrase. In the introduction and inmate conclusions.
Response: In order to respond to the reviewer 2 comments, we have changed the Abstract section; trying to clarify it.
Reviewer 2: 30 teeth or 42? Is not clear.
Response: In order to respond to the reviewer 2 comments, we clarify that 3 experimental models with 14 teeth each (from tooth 1.7 to tooth 2.7) were manufactured, but in the study the brackets were cemented from the tooth 1.5 to the tooth 2.5 (it is indicated in line 111).
Reviewer 2: No mention about integrity of selected teeth was underlined.
Response: In order to respond to the reviewer 2 comments, we have described the integrity of selected teeth.
Reviewer 2: Who removed the remain composite from teeth? Which handpiece? With or without water? Speed? One bur for all teeth? Same operator? Which force applied?
Response: In order to respond to the reviewer 2 comments, we clarify the protocol used and the manuscript was modified: “The remains of composite resin cement were removed with gently removed in a unidirectional movement by a single operator, using a low-speed contra angle handpiece, (W&H WE-99 LED G, Bürmoos, Austria), at 1500rpm with profuse irrigation. A polishing diamond bur surface (Ref. 882 314 012, Komet Medical, Lemgo, Germany) was used for each experimental model, until there were not clinically visible”
Reviewer 2: It's not clear why 2 alignments were required.
Response: In order to respond to the reviewer 2 comments, we clarify that the first alignment could be less accurate because the mesh is bigger. When we segment each single tooth we reduce the mesh in multiple smaller meshes and therefore we obtained better accuracy in the alignment of those smaller meshes. A modification of the manuscript was done: “There are several factors which may affect the digital impression accuracy. One of those factors is the distance from the first point scanned, used as reference, and the succeeding points in the scanned structures, which would have been stitched to the previous one. Each individual stitch represents a chance to incurring an error”. Furthermore, the segmentation and realignment would be strictly necessary for future in vivo investigation where a tooth would be in a different position before and after the orthodontic treatment, hence the segmentation and realignment would be necessary.
Reviewer 2: It's not clear how many teeth were evaluated in total (all the teeth?).
Response: In order to respond to the reviewer 2 comments, we have described in line 111 that 30 teeth were analyzed and it can also be seen in table 1 of the Results section.
Reviewer 2: Why a diamond bur was used to eliminate the cement? Literature reports other kind of burs for this procedure.
Response: In order to respond to the reviewer 2 comments, we clarify that the authors decided to use a diamond bur to obtain a higher polish to compare; however, the analysis of the cement removal material is not the objective of this study, but this new measurement technique will allow its analysis in future studies: “in order to analyze if it was possible to measure the volume of enamel removed, the authors used a “more aggressive” method. If a very conservative method would be used, an untouched enamel could be obtained, so volume and area of the enamel could not been calculated and consequently the repeatability and reproducibility could not been analyzed.
Reviewer 2: Validation: any test was performed?
Response: In order to respond to the reviewer 2 comments, we clarify that the validation of the measurement technique was performed with the Gage R&R statistical analysis; it is indicated in line 274.
We take this opportunity to thank the recommendations and suggestions made by you to improve the document.
Yours sincerely,
Reviewer 3 Report
In my opinion this study has a purely academic and research interest, as the evaluation is carried out in-vitro and does not find great applications from the clinical point of view. However, the study is well prepared and with a correct scientific methodology and could be the starting point for further investigations of this type in vivo. I suggest, however, to provide more details in reproducibility as regards the phase of removing the brackets, as the authors illustrate only the type of device used. I also recommend reducing the number of figures and images to make the text smoother. We also advise authors to read this article "Effects of stannous fluoride on eroded enamel permeability.J Biol Regul Homeost Agents. 2018 Mar-Apr; 32 (2 Suppl. 2): 1-8.", which provides a possible treatment for those dental elements that have been damaged also by an inattentive debonding procedure.
Author Response
Dear Reviewer 3,
I’m pleased to resubmit the manuscript of the work entitled, “Novel digital technique to quantify the area and volume of cement remaining and enamel removed after fixed multibracket appliance therapy debonding: An in vitro study”.
I will respond to your comments and suggestions below:
Reviewer 3: English language and style are fine/minor spell check required.
Response: In order to adapt to the reviewer 3 comments, we have sent the manuscript to the English editing service of MDPI editorial (We attached the English Editing Certificate).
Reviewer 3: I suggest, however, to provide more details in reproducibility as regards the phase of removing the brackets, as the authors illustrate only the type of device used.
Response: In order to respond to the reviewer 3 comments, we have described the protocol used to remove multibracket appliance and the cement remaining (line 117-125):
“Then, the fixed multibracket appliance was removed from tooth 1.5 to 2.5 with a specific instrument to remove the fixed multibracket appliance (MBT, .022, Pacific Orthodontics, Guadalajara, Spain) (Figure 1C) and a post-removing brackets intraoral scan (True Definition, 3M ESPE ™) (STL3) was performed (Figure 2C). The remains of composite resin cement were gently removed in a unidirectional movement by a single operator, using a low-speed contra angle handpiece, (W&H WE-99 LED G, Bürmoos, Austria), at 1500rpm with profuse irrigation. A polishing diamond bur surface (Ref. 882 314 012, Komet Medical, Lemgo, Germany) was used for each experimental model. until there were not clinically visible (Figure 1D) and a post-removing cement intraoral scan (True Definition, 3M ESPE ™) (STL4) was performed (Figure 2D)”
Reviewer 3: I also recommend reducing the number of figures and images to make the text smoother.
Response: In order to respond to the reviewer 3 comments, we clarify that this novel, repeatable, reproducible, and accurate measurement procedure may serve for future studies related to dentistry or other disciplines, and we also believe that it is necessary to illustrate the procedure so that future researchers can reproduce the measurement procedure.
Reviewer 3: We also advise authors to read this article "Effects of stannous fluoride on eroded enamel permeability.J Biol Regul Homeost Agents. 2018 Mar-Apr; 32 (2 Suppl. 2): 1-8.", which provides a possible treatment for those dental elements that have been damaged also by an inattentive debonding procedure.
Response: In order to respond to the reviewer 3 comments, we have included the reference in the Introduction section.
We take this opportunity to thank the recommendations and suggestions made by you to improve the document.
Yours sincerely,
Round 2
Reviewer 3 Report
I think that the authors have done a good job and that they made all the corrections suggested